# High Serum Levels of Wnt Signaling Antagonist Dickkopf-Related Protein 1 Are Associated with Impaired Overall Survival and Recurrence in Esophageal Cancer Patients

**DOI:** 10.3390/cancers13194980

**Published:** 2021-10-04

**Authors:** José Giron Ramirez, Daniel J. Smit, Fabrice Viol, Jörg Schrader, Tarik Ghadban, Klaus Pantel, Jakob R. Izbicki, Matthias Reeh

**Affiliations:** 1Department of General, Visceral and Thoracic Surgery, University Medical Center Hamburg-Eppendorf, 20246 Hamburg, Germany; j.giron-ramirez@uke.de (J.G.R.); jschrader@uke.de (J.S.); t.ghadban@uke.de (T.G.); izbicki@uke.de (J.R.I.); 2Institute of Biochemistry and Signal Transduction, University Medical Center Hamburg-Eppendorf, 20246 Hamburg, Germany; d.smit@uke.de; 3I. Department of Medicine, University Medical Center Hamburg-Eppendorf, 20246 Hamburg, Germany; f.viol@uke.de; 4Department of Tumor Biology, University Medical Center Hamburg-Eppendorf, 20246 Hamburg, Germany; pantel@uke.de

**Keywords:** esophageal carcinoma, DKK1, prognostic marker, biomarker, precision medicine, circulating tumor cells, disseminated tumor cells

## Abstract

**Simple Summary:**

Dickkopf-related protein 1 (DKK1), an antagonist of the canonical Wnt pathway has been the subject of research for many years. Especially in gastrointestinal cancers, research suggests a pivotal role of DKK1. In order to understand the role of DKK1 in esophageal cancer, we analyzed blood samples of esophageal cancer patients for their DKK1 levels and retrospectively analyzed the clinicopathological data. In our study cohort, we observed a negative prognostic role of high DKK1 serum levels with respect to overall survival in esophageal cancer patients. These data may suggest serum DKK1 as a novel biomarker for improved risk stratification and treatment monitoring in esophageal cancer patients.

**Abstract:**

Dickkopf-related protein 1 (DKK1), an antagonist of the canonical Wnt pathway, has received tremendous attention over the past years as its dysregulation is said to be critically involved in a wide variety of gastrointestinal cancers. However, the potential clinical implications of DKK1 remain poorly understood. Although multimodal treatment options have been implemented over the past years, esophageal cancer (EC) patients still suffer from poor five-year overall survival rates ranging from 15% to 25%. Especially prognostic factors and biomarkers for risk stratification are lacking to choose the most beneficial treatment out of the emerging landscape of different treatment options. In this study, we analyzed the serum DKK1 (S-DKK1) levels of 91 EC patients prior to surgery in a single center study at the University Medical Center Hamburg-Eppendorf by enzyme-linked immunosorbent assay. High levels of S-DKK1 could be especially observed in patients suffering from esophageal adenocarcinoma which may promote the hypothesis of a crucial role of DKK1 in inflammation. S-DKK1 levels of ≥5800 pg/mL were shown to be associated with unfavorable five-year survival rates and the presence of CTCs. Interestingly, significantly lower S-DKK1 levels were detected in patients after neoadjuvant treatment, implying that S-DKK1 may serve as a useful biomarker for treatment monitoring. Multivariate analysis identified S-DKK1 as an independent prognostic marker with respect to overall survival in EC patients with a hazard ratio of 2.23. In conclusion, our data implicate a negative prognostic role of DKK1 with respect to the clinical outcome in EC patients. Further prospective studies should be conducted to implement S-DKK1 into the clinical routine for risk stratification and treatment monitoring.

## 1. Introduction

Esophageal cancer (EC) is a devastating malignant disease with half a million deaths per year worldwide [1]. Two subsets, namely, esophageal squamous cell carcinoma (ESCC) and esophageal adenocarcinoma (EAC), mainly dominate the histopathological landscape of esophageal tumors [2]. However, the pathogenesis of these two entities greatly differs. ESCC mostly develops in the proximal esophagus due to exposition to exogenous toxics including tobacco, alcohol and radiation. In contrast, EAC develops as a metaplasia due to gastroesophageal reflux with subsequent development of dysplastic adenoma and transformation to malignant adenocarcinoma [2]. Curative treatment of EC includes surgery, chemotherapy and/or radiochemotherapy, either in a neoadjuvant or adjuvant treatment regimen. Despite the broad range of cytotoxic agents and the advances in surgery and radiotherapy, patients still have poor five-year overall survival (OS) rates of 15% to 25% [3]. 

The canonical (ß-catenin dependent) Wnt signaling pathway plays a crucial role in embryonic development and maintenance of stemness [4]. Aberrant activity of the Wnt signaling pathway has been shown to play a pivotal role in tumor development by regulating cell proliferation [5,6]. Dickkopf family member Dickkopf-related protein 1 (DKK1) acts as a Wnt signaling pathway antagonist by interacting with low-density lipoprotein receptor-related protein 6 (LRP6) co-receptor. Thereby, it prevents the complex formation of LRP6, Wnt and frizzled receptor leading to internalization and inactivation of the downstream ß-catenin pathway [7]. Nevertheless, pleiotropic roles for Wnt signal pathway have been reported in the past [8]. As DKK1 is a potent inhibitor of Wnt signaling it was originally characterized as a tumor suppressor, however, recent research identified DKK1 expression as a negative prognostic marker in several cancer entities [8,9]. For example, Yamabuki et al. analyzed primary tumor samples of EC patients by immunohistochemistry and were able to identify DKK1 expression as a negative prognostic marker for OS in these patients [10]. However, this study did not particularly focus on the two major histological subtypes. 

The aim of this study was to determine the impact of S-DKK1 on clinicopathological parameters, neoadjuvant treatment, recurrence of cancer and overall survival in EC patients. For this purpose, a cohort of 91 patients suffering from EC was retrospectively analyzed and the findings related to their S-DKK1 levels. 

## 2. Materials and Methods

### 2.1. Study Population

The study includes a total of 91 patients suffering from esophageal cancer (30 squamous cell carcinoma and 61 adenocarcinoma patients, respectively) who underwent surgery with curative intention at the Department of General-, Visceral- and Thoracic Surgery at the University Medical Center Hamburg-Eppendorf between 2008 and 2011. The study was approved by the Medical Ethical Committee, Hamburg, Germany and complies with the principles of the Declaration of Helsinki. Informed consent was obtained from all patients. The study cohort consists of 65 men (71.4%) and 26 women (28.6%). The median age of the esophageal cancer patients was 65 years. All tumors were histopathologically confirmed and tumor stages encoded according to 8th edition of the UICC TNM Classification of Malignant Tumors.

All patients were treated according to national guidelines. Briefly, patients were treated stage-dependent either by primary resection of the tumor or by neoadjuvant chemotherapy/radiochemotherapy followed by surgical removal of the tumor. A group of 30 obesity WHO grade III (body-mass-index > 40 kg/m^2^) patients served as a control group. All obesity patients underwent esophagogastroduodenoscopy (<6 months) prior to bariatric surgery. None of these patients had a history of esophageal carcinoma, a present esophageal carcinoma nor suffered from metaplasia of the distal esophagus. One third of the obesity patients were male (33.3%) and two third (66.6%) were female. The median age within this group was 44 years.

### 2.2. Acquisition of Clinical and Pathological Data

All data were extracted from the clinical records of the University Medical Center Hamburg-Eppendorf after written informed consent was obtained. 

### 2.3. Enzyme-Linked Immunosorbent Assay (ELISA) for DKK1 Levels

Blood samples of patients suffering from esophageal cancer were collected prior to surgical removal of the primary tumor, separated by centrifugation and serum samples stored at −80 °C. The serum DKK1 (S-DKK1) levels were determined using the DKK1 Human ELISA Kit (#EHDKK1, ThermoFisher Scientific, Waltham, MA, USA). In brief, the samples were thawed on ice, diluted according to the manufacturer’s instructions and transferred to a microtiter plate with precoated DKK1-specific monoclonal antibodies. After incubation, the wells were thoroughly washed, incubated with a second antibody against DKK1 and incubated again. Thereafter, the wells were washed, and a biotin-conjugated antibody was added to the microtiter plate. S-DKK1 levels were determined after addition of streptavidin horseradish peroxidase conjugate and adsorption measured photometrically at 450 nanometer and 550 nanometer wavelengths in a microplate reader (FLUOstar Omega, BMG Labtech, Ortenberg, Germany). All samples were analyzed in duplicates and a standard curve of supplied recombinant DKK1 standard was created once per assay. Absorption values at 550 nm were subtracted prior to further analysis. S-DKK1 concentrations (pg/mL) of the patient samples were calculated according to the formula of the standard curve. 

### 2.4. Detection of Circulating Tumor Cells

Circulating tumor cell (CTC) analysis was performed using the CellSearch system as previously described [11]. Blood samples (7.5 mL) were collected in CellSave preservative tubes, stored at room temperature and processed within 48 hours, according to the manufacturer’s instructions. The accuracy and reproducibility of the CellSearch system has been described previously [11,12]. Presence of a nucleus, cytokeratin expression, round or oval cell morphology and absence of CD45 expression were the criteria for CTCs. The cut-off value for CTC positivity was 1 CTC/7.5 mL [13].

### 2.5. Detection of Disseminated Tumor Cells

Bone marrow aspiration from the upper iliac crest, enrichment of mononuclear cells by Ficoll density gradient centrifugation, preparation of cytospins and immunostaining were performed as described elsewhere [14,15,16]. Nucleated cells, which expressed the cytokeratins 8, 18 and 19, detected by the pan anti-keratin antibody A45-B/B3 (mouse IgG1, AS Diagnostics) were identified as DTCs. The criteria for DTC categorization were described elsewhere [15,16,17].

### 2.6. Statistical Analysis

Statistical analysis of the data was carried out using SPSS Statistics version 27 (IBM Inc., Armonk, NY, USA). Missing data sets were defined prior to analysis and excluded. Cross tables were plotted, and the statistical significance tested with two-sided chi squared/fisher exact test. A *p*-value of <0.05 was considered as statistically significant. To assess the statistical significance of two groups, Levene test for variance equality followed by either unpaired two-sided *t*-test or unpaired two-sided *t*-test with Welch correction was applied, where appropriate. 

For overall survival and recurrence analysis Kaplan–Meier curves were plotted and univariate analysis carried out by the logrank test (Mantel–Cox). Patients who were lost during follow up were censored at their last documented visit. Multivariate analysis was conducted using the Cox-regression method. A *p*-value of <0.05 was considered as statistically significant.

## 3. Results

### 3.1. S-DKK1 Levels in Esophageal Cancer Patients

As the use of serum DKK1 (S-DKK1) is limited to research and not yet established in the clinical setting, we determined the mean S-DKK1 of our 91 patients suffering from esophageal cancer (EC) and 30 obesity patients who served as the healthy control before surgery. In obese patients, mean S-DKK1 levels of 3910 pg/mL ± 354.10 standard error of the mean (SEM) and a minimum of 1440 pg/mL and a maximum of 8840 pg/mL were measured. The mean S-DKK1 levels were higher in patients suffering from EC. S-DKK1 levels ranged from 540 pg/mL to 62,930 pg/mL with a mean of 5989.60 pg/mL ± 871.33 SEM in our EC study cohort. In esophageal adenocarcinoma (EAC) patients, a significantly higher S-DKK1 with a mean of 7395.14 pg/mL ± 1245.78 SEM was found compared to the healthy controls (*p* = 0.009). The S-DKK1 levels in esophageal squamous cell carcinoma (ESCC) did not significantly differ from the healthy control group with a mean of 3131.67 pg/mL ± 444.87 SEM (*p* = 0.176) (Figure 1).

Based on the mean value of S-DKK1 in all cancer patients, the cancer patients were divided in two subgroups with low S-DKK1 (<5800 pg/mL) and high S-DKK1 (≥5800 pg/mL) (Table 1). The S-DKK1 of the patients was not significantly altered by age and gender in EC patients (*p* = 0.820 and *p* = 0.208, respectively). However, in patients suffering from EAC, more than one third (39.3%) of all patients had S-DKK1 levels ≥ 5800 pg/mL compared to ESCC patients (13.3%) (*p* = 0.015). Additionally, significantly different S-DKK1 levels were observed with respect to tumor stage (*p* = 0.047) and lymphatic vessel infiltration (*p* = 0.093), while no significant differences were observed for lymphatic node metastasis (*p* = 0.198), distant metastasis (*p* = 0.550), venous vessel infiltration (*p* = 0.108), tumor grading (*p* = 0.553) and resection status of the tumor margins (*p* = 0.542). Fifteen patients (16.5%) received neoadjuvant treatment prior to surgery using chemotherapy or radio-chemotherapy. Interestingly, S-DKK1 levels were significantly lower after neoadjuvant treatment (*p* = 0.031). Circulating tumor cells (CTCs) and disseminated tumor cells (DTCs) were more frequently observed in EC patients with high S-DKK1 (*p* ≤ 0.001 and *p* = 0.003, respectively). In ESCC patients, the recurrence rates (*p* = 0.037) were significantly altered with respect to the S-DKK1 levels. Remarkably, all patients who did not relapse on their ESCC were in the low S-DKK1 group (Table 2). In EAC patients, the S-DKK1 levels also significantly altered the recurrence rates (*p* = 0.029) and, additionally, the presence of CTCs (*p* = 0.004) and the presence of DTCs (*p* = 0.013) (Table 3). 

### 3.2. Recurrence of Cancer Occurs Earlier in Patients with High S-DKK1 and Detectable CTCs

To further evaluate the impact of S-DKK-1, the presence of CTCs and DTCs patients were followed up until the recurrence of the EC. A univariate analysis using the log rank test revealed that patients with high S-DKK1 relapse earlier within a median of 12 months (95% CI 8.2–15.8 ± S.E. 1.94), compared to 22 months (95% CI 15.43–28.57 ± S.E. 3.35) in patients with low S-DKK1 (*p* < 0.0005) (Figure 2A). Furthermore, we were able to detect that the absence of CTCs in the blood is favorable with respect to recurrence in our EC cohort (Figure 2B). Patients with detectable CTCs tend to relapse earlier after a median of 11 months (95% CI 9.01–12.99 ± S.E. 1.02) compared to a time to relapse of 22 months (95% CI 12.43–31.57 ± S.E. 4.88) in patients with no detectable CTCs (*p* = 0.003) (Figure 2B). No significant difference could be detected regarding the presence of DTCs in EC relapse in our study cohort (Figure 2C). However, the data indicate a trend that the absence of DTCs, analogue to the absence of CTCs, leads to a favorable effect on the recurrence-free survival (median 18 months versus 11 months, *p* = 0.059) (95% CI 8.3–27.70 ± S.E. 4.95 and 95% CI 7.33–14.67 ± S.E. 1.87, respectively) (Figure 2C, Table 4). 

In EAC, a strong trend of earlier recurrence could be observed in the high S-DKK1 group with a median of 13 months (95% CI 6.96–19.04 ± S.E. 3.08) compared to patients with low S-DKK1 with a median of 17 months (95% CI 9.95–24.05 ± S.E. 3.60) (*p* = 0.064) (Figure 3A). Whereas SCC patients with high levels of S-DKK1 showed significant shorter recurrence-free survival compared to SCC patients with low S-DKK1 levels (*p* < 0.0001) (Figure 3B).

### 3.3. High Levels of S-DKK1 and the Presence of CTCs in Blood Are Associated with Lower Overall Survival in Esophageal Cancer Patients

The Kaplan–Meier survival analysis revealed that EC patients with high S-DKK-1 levels have an impaired overall survival (OS) of a median of 14 months (95% CI 8.65–19.36 ± S.E. 2.73) compared to 26 months in patients with low S-DKK (95% CI 21.30–30.70 ± S.E. 2.40) (*p* = 0.003) (Figure 4A). Furthermore, patients with CTCs present in the blood stream had a lower OS of a median of 13 months (95% CI 6.59–19.41 ± S.E. 3.27) compared to 26 months (95% CI 21.47–30.53 ± S.E. 2.31) in patients with no detectable CTCs (*p* = 0.010) (Figure 4B). 

There was no significant difference, but only a trend could be observed in patients with detectable DTCs in their bone marrow. Patients suffering from EC with detectable DTCs had a median OS of 13 months (95% CI 5.67-20.33 ± S.E. 3.74) compared to 24 months with no DTCs present (95% CI 17.58–30.42 ± S.E. 3.28) (*p* = 0.116) (Figure 4C). A summary of the data can be found in Table 5. Multivariate analysis by Cox-regression revealed a hazard ratio of 2.23 (95% CI 1.19–4.17 ± S.E. 0.32) for death in the high S-DKK1 group compared to the low S-DKK1 group (*p* = 0.012). Additionally, the presence of lymph node metastasis has been demonstrated as an independent prognostic factor in EC patients by multivariate analysis (*p* = 0.024). Other well-established clinicopathological parameters including tumor stage (*p* = 0.157) and presence of distant metastasis (*p* = 0.179) as well as age group (*p* = 0.156) and gender (*p* = 0.555) did not significantly alter the risk of death in our study cohort. 

Next, we analyzed the overall survival within the two histological subgroups (ESCC and EAC). EAC patients with low S-DKK1 levels had a higher, but not significant, median overall survival of 25 months (95% CI 20.11–29.89 ± S.E. 2.49) compared to patients with high serum levels of DKK1 with a median of 14 months (95% CI 9.08–18.93 ± S.E. 2.51) (*p* = 0.062) (Figure 5A). Similarly, in ESCC patients, high S-DKK1 levels led to a trend of impaired overall survival with a median of 9 months (95% CI 0–29.58 ± S.E. 10.50) compared to 44 months (95% CI 9.27–78.73 ± S.E. 17.72) in the low S-DKK1 group (*p* = 0.066) (Figure 5B). Although the univariate analysis by the log rank test revealed that the overall survival in the subgroups is not statistically significant, it rather represents a strong trend for the impact of S-DDK1 on overall survival in EAC and ESCC. 

## 4. Discussion

Esophageal cancer patients suffer from the sixth highest mortality among all cancer entities [1]. Despite the fact that multimodal therapy approaches have been implemented and improved over the past years, patients still suffer from poor 5-year survival rates. Prognostic factors that determine the outcome of EC patients and thereby allow accurate risk stratification are especially lacking. In this work, we provide evidence that S-DKK1 may serve as a useful biomarker in esophageal cancer patients. DKK1, as a negative regulator of canonical Wnt signaling, has been an emerging topic of experimental as well as clinical research for cancer patients [5]. DKK1 has not only been evaluated as a prognostic marker in a wide variety of cancers [9] but may also serve as a potential novel treatment target [18]. Recently, Goyal et al. reported that DKN-01, an inhibitor of DKK-1, was well tolerable in a phase I clinical trial in combination with gemcitabine or cisplatin in patients with advanced cancer of the biliary tract [19]. 

Yamabuki et al. were the first to suggest a crucial role of DKK1 in lung cancer and esophageal cancer [10]. The authors demonstrated a strong upregulation of DKK1 in various lung cancer and esophageal cancer cell lines. Of the 280 primary EC tissues analyzed, more than 60% were stained positive for DKK1 [10]. Furthermore, the authors were able to demonstrate a significant unfavorable impact on overall survival of high DKK1 expression in surgically treated lung cancer and EC samples [10]. In line with the data from this previous study on the role of DKK-1 in EC, we observed an unfavorable impact of high S-DKK1 levels with respect to overall survival. However, the authors did only analyze samples from ESCC patients and not EAC patients [10]. Therefore, we were able to extend the current knowledge on overall survival, recurrence and the presence of CTCs and DTCs for the two most common histopathological subtypes of EC. CTC and DTC detection in cancer patients is a major topic in today’s cancer research and provides a rationale for risk stratification [20,21,22], treatment monitoring [23] and novel treatment strategies for personalized medicine [24,25,26,27]. In the past, we reported that CTCs are an independent prognostic indicator of overall survival and tumor recurrence and thus may improve preoperative staging [20]. In our study cohort, CTCs as well as DTCs were more prevalent in patients with high S-DKK1 levels compared to patients with low S-DKK1 levels, which may underline the high significance of DKK1 in EC pathogenesis and malignancy. 

Despite extensive research in the past, in contrast to other gastrointestinal malignancies, including colorectal cancer, no established tumor markers for esophageal cancer exist yet [28]. Nevertheless, a meta-analysis by Zhang et al. demonstrated that tumor markers including carcinoembryonic antigen, cytokeratin-19 fragment 21-1, p53 antibody, squamous cell carcinoma antigen and VEGF-C are highly specific but lack the required sensitivity to precisely diagnose EC [29]. Therefore, multiple biomarkers, including S-DKK1, may be combined to increase sensitivity. 

Due to the growing diversity and complexity in treatment selection (e.g., chemotherapeutics, targeted therapies, small molecule inhibitors and radiotherapy) within the multimodal approach, S-DDK1 may complement already established parameters for decision making. The implementation of S-DKK1 into the clinical routine may also facilitate the identification of those patients who could benefit from neoadjuvant or adjuvant treatment, as we were able to detect significantly lower S-DKK1 levels in patients who underwent neoadjuvant treatment. However, uniform reference ranges for S-DKK1 determined by a large cohort screening are still not established. 

Another potential application may be S-DKK1 as a novel treatment target. Targeted therapies in EC have recently been established against EC with distinct molecular signatures [30]. For example, trastuzumab, a monoclonal antibody targeting human epidermal growth factor receptor 2 (HER2) in combination with chemotherapy, has recently been approved as first line therapy by the FDA for EC patients [30]. This drug targets the constitutive activation of several pathways including the oncogenic RAS/RAF/MEK/ERK pathway, which is subsequently activated by the amplification of HER2 [31]. However, as discussed above, the functional role of DKK1 is still unknown, and therefore targeted therapy inhibiting DKK1 should be carefully evaluated in further experiments in vitro as well as xenograft models in vivo. Recently, Lyros et al. reported that DKK1 promotes tumor growth by attenuating the PI3K/AKT/mTOR axis independently of Wnt signaling in EC OE33 cells in vitro [32]. While according to this study an inhibition of DKK1 would be beneficial, in other studies, including breast cancer, a tumor-suppressive effect of DKK1 with respect to migration and invasion was observed [33]. In our study, we found out that high levels of S-DKK1 are associated with poor prognosis. Therefore, the data are rather in line with a tumorigenic role of DKK-1, although further molecular analysis is necessary to confirm this idea. In primary prostate cancer, a strong positive correlation of S-DKK1 levels and tissue DKK1 expression levels within the tumor was reported in the past [34]. Nevertheless, it remains unclear whether the high S-DKK1 levels are a direct tumorigenic stimulus that promotes tumor growth and malignancy or whether they are induced by a negative feedback loop as a sign of constitutive Wnt signaling. Hence, further molecular characterization of the underlaying mechanism is needed to provide a robust answer to the question whether DKK1 may be a suitable target in EC.

In addition to treatment choice and the potential use as a treatment target, the use of S-DKK1 in EC may also facilitate the identification of high-risk patients that would benefit from intensified aftercare (e.g., in the form of additional clinical imaging and shorter consultation intervals). Recent research demonstrated that in lung cancer, especially, the combination of blood-based biomarkers and CTC detection is superior to use of single biomarkers [35]. Therefore, we think that the combination of CTCs and S-DKK1 levels may be useful for a more personalized cancer therapy. 

Additionally, the emerging importance of neoadjuvant and adjuvant therapy regimens in the past decade led to a paradigm change in EC [36,37]. In our study cohort, which includes patients from the years 2008 to 2011, only 16.5% received neoadjuvant treatment prior to surgery. Therefore, new prospective studies should be started to evaluate the impact of neoadjuvant treatment on S-DKK1 in EC patients. The pathogenesis of ESCC and EAC differs considerably and therefore a further analysis of the role of DKK-1 in the subtypes of EC is also necessary. Gastroesophageal reflux is a major driver of esophageal Barrett metaplasia and predisposes as a precancerous disease for the development of EAC. While the functional role of DKK1 in cancer progression has been evaluated frequently in the past [38], the impact on the pathogenesis of EAC remains poorly understood. 

Studies suggest a critical influence of the inflammatory microenvironment due to a long history of reflux [39,40]. Upregulation of DKK1 and consecutive dysregulation of the Wnt signaling pathway due to gastric acid mediated damage was observed in vitro [41]. In addition, Darlavoix et al. reported that elevated DKK1 expression is associated with a higher risk of malignant progression of Barrett’s metaplasia [42]. In our study, EAC cancer patients had significantly higher S-DKK1 levels compared to a healthy control with no history of EC. Additionally, we observed high levels of S-DKK (≥5800 pg/mL) in almost 40% of our EAC patients compared to 13% of our ESCC patients which may underline the proposed mechanism of inflammation-induced carcinogenesis by DKK1 in EAC. Univariate analysis of OS and time to recurrence revealed a strong trend of impaired OS and earlier recurrence within the EAC subgroup. Therefore, this work provides a rationale for a pivotal role of S-DKK1 in carcinogenesis, inflammation and clinical treatment monitoring, which should be further evaluated in prospective clinical studies. 

## 5. Conclusions

In this study, we retrospectively analyzed the impact of S-DKK1 levels in 30 esophageal squamous cell cancer and 61 esophageal adenocarcinoma patients. S-DKK1 levels were significantly higher in esophageal adenocarcinoma patients compared to healthy control. Interestingly, significantly lower S-DKK1 levels were observed after neoadjuvant treatment, indicating that S-DKK1 may serve as a novel biomarker for treatment response in esophageal cancer. Additionally, high levels of S-DKK1 were associated with unfavorable overall survival rates and a shorter time until the recurrence of the esophageal cancer. The presence of CTCs and DTCs was also significantly enhanced in patients with high S-DKK1 levels (≥5800 pg/mL). Multivariate analysis by cox regression revealed that high S-DKK1 is an independent prognostic marker in patients with esophageal cancer. In conclusion, these data suggest a crucial role for DKK1 in esophageal carcinoma as a biomarker for treatment response and poor overall survival, which should be further evaluated in prospective studies to be implemented into clinical routine. Moreover, our results encourage future studies on DKK1 as a potential therapeutic target in EC.

## Figures and Tables

**Figure 1 cancers-13-04980-f001:**
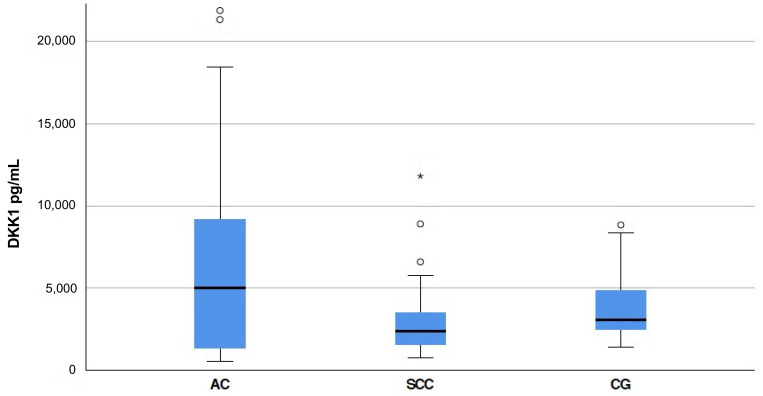
S-DKK1 levels in the esophageal cancer study cohort analyzed. Boxplot of S-DKK1 levels in esophageal cancer patients with standard deviation determined by ELISA. The esophageal adenocarcinoma (AC) group consists of 61 patients, the esophageal squamous cell carcinoma (SCC) patients’ group and the healthy control (CG) consist of 30 patients each. Extreme values were marked in the Figure and encoded as following: ᵒ, > 1.5 box lengths from one hinge of the box; *, > 3 box lengths from a hinge of the box. Each symbol represents the S-DKK1 level of one patient.

**Figure 2 cancers-13-04980-f002:**
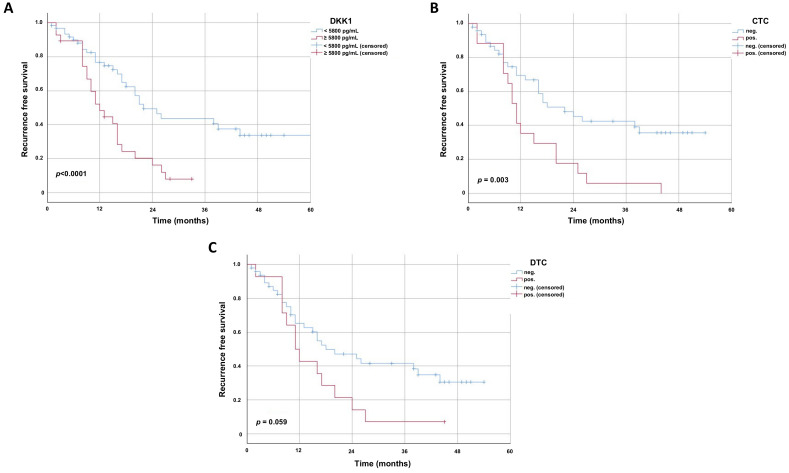
Kaplan–Meier survival curves displaying time to recurrence in 91 esophageal cancer patients. Kaplan–Meier survival curves displaying time to recurrence in months of esophageal cancer patients (*n* = 91). Univariate analysis was carried out by the logrank test (Mantel–Cox). A *p*-value < 0.05 was considered as statistically significant. Patients who were lost during follow up were censored at their last documented visit. (**A**) Time to recurrence of esophageal cancer patients with respect to S-DKK1 levels. Subgroups were built based on the mean values of all S-DKK1 levels in the 91 esophageal cancer patients. (**B**) Time to recurrence of esophageal cancer patients with respect to the presence of CTCs in the blood. A cut-off value of at least one CTC was used to define the presence of CTCs. (**C**) Time to recurrence of esophageal cancer patients with respect to the presence of DTCs in the blood. A cut off value of at least one DTC was used to define the presence of DTCs.

**Figure 3 cancers-13-04980-f003:**
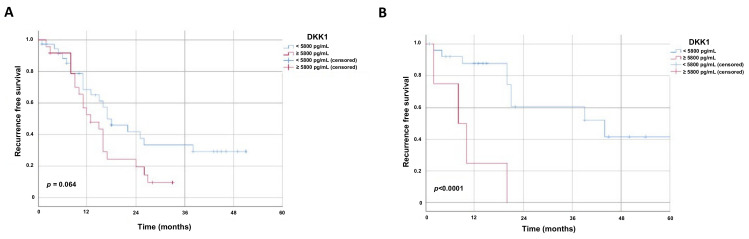
Kaplan–Meier survival curves displaying time to recurrence in esophageal carcinoma patients. Kaplan–Meier survival curves displaying time to recurrence in months within the (**A**) esophageal adenocarcinoma and (**B**) esophageal squamous cell cancer patients. Univariate analysis was carried out by the logrank test (Mantel–Cox). A *p*-value < 0.05 was considered as statistically significant. Patients who were lost during follow up were censored at their last documented visit. Time to recurrence of esophageal cancer patients with respect to S-DKK1 levels is displayed. Subgroups were built based on the mean values of all S-DKK1 levels in the 91 esophageal cancer patients (ESCC and EAC).

**Figure 4 cancers-13-04980-f004:**
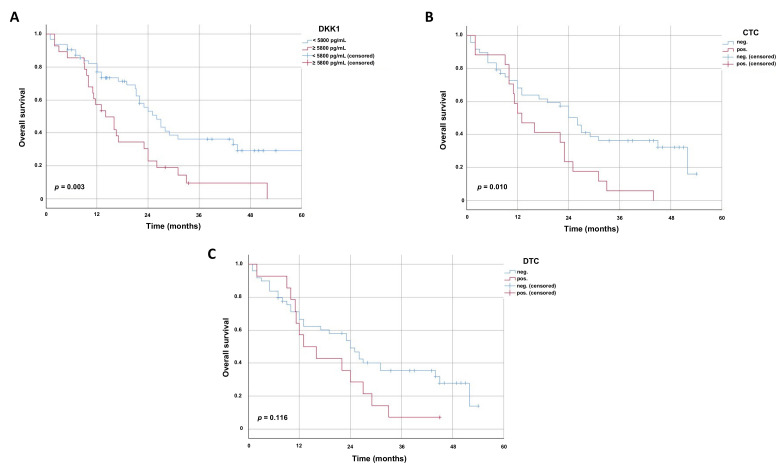
Kaplan–Meier survival curves displaying overall survival of 91 esophageal cancer patients. Kaplan–Meier survival curves displaying overall survival in months of esophageal cancer patients (*n* = 91). Univariate analysis was carried out by logrank test (Mantel–Cox). A *p*-value < 0.05 was considered as statistically significant. Patients who were lost during follow up were censored at their last documented visit. (**A**) Overall survival of esophageal cancer patients with respect to S-DKK1 levels. Subgroups were built based on the mean values of all S-DKK1 levels in the 91 esophageal cancer patients. (**B**) Overall survival of esophageal cancer patients with respect to the presence of CTCs in the blood. A cut off value of at least one CTC was used to define the presence of CTCs. (**C**) Overall survival of esophageal cancer patients with respect to the presence of DTCs in the blood. A cut off value of at least one DTC was used to define the presence of DTCs.

**Figure 5 cancers-13-04980-f005:**
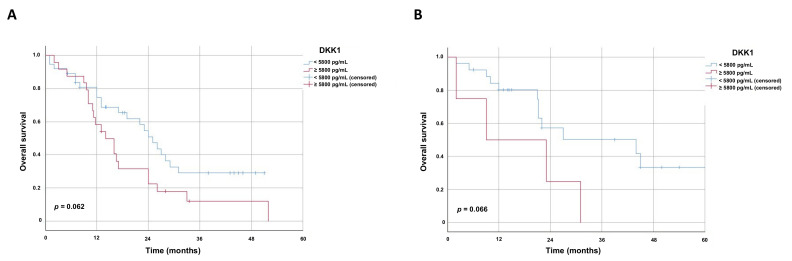
Kaplan–Meier survival curves displaying overall survival of esophageal squamous cell and esophageal adenocarcinoma patients. Kaplan–Meier survival curves displaying the overall survival in months within the (**A**) esophageal adenocarcinoma and (**B**) esophageal squamous cell cancer patients. Univariate analysis was carried out by the logrank test (Mantel–Cox). A *p* value < 0.05 was considered as statistically significant. Patients who were lost during follow up were censored at their last documented visit. Time to recurrence of esophageal cancer patients with respect to S-DKK1 levels is displayed. Subgroups were built based on the mean values of all S-DKK1 levels in the 91 esophageal cancer patients (ESCC and EAC).

**Table 1 cancers-13-04980-t001:** Association of S-DKK1 with clinicopathological parameters in esophageal cancer.

		*N* (%)	Low S-DKK1 <5800 pg/mL *N* (%)	High S-DKK1 ≥5800 pg/mL *N* (%)	*p* Value (a) Two-Sided Fisher Exact Test (b) Two-Sided Pearson Chi Quadrat Test
**Gender**					0.208 (a)
	Female	26 (28.6%)	21 (80.8%)	5 (19.2%)	
	Male	65 (71.4%)	42 (64.6%)	23 (35.4%)	
**Age**					0.820 (a)
	<65	42 (46.2%)	30 (71.4%)	12 (28.6%)	
	≥65	49 (53.8%)	33(67.3%)	16 (32.7%)	
**Tumor Entity**					0.015 (a)
	Squamous cell carcinoma	30 (33%)	26 (86.7%)	4 (13.3%)	
	Adenocarcinoma	61 (67%)	37 (60.7%)	24 (39.3%)	
**Tumor Stage**					0.047 (b)
	T0	2 (2.2%)	2 (100%)	0 (0%)	
	T1	21 (23.1%)	17 (81%)	4 (19%)	
	T2	14 (15.4%)	13 (92.9%)	1 (7.1%)	
	T3	48 (52.7%)	28 (58.3%)	20 (41.7%)	
	T4	6 (6.6%)	3 (50%)	3 (50%)	
**Lymphatic Node Metastasis**					0.198 (b)
	N0	34 (37.4%)	27 (79.4%)	7 (20.6%)	
	N1	28 (30.8%)	16 (57.1%)	12 (42.9%)	
	N2	17 (18.7%)	13 (76.5%)	4 (23.5%)	
	N3	12 (13.2%)	7 (58.3%)	5 (41.7%)	
**Distant Metastasis**					0.550 (a)
	M0	88 (96.7%)	60 (68.2%)	28 (31.8%)	
	M1	3 (3.3%)	3 (100%)	0 (0%)	
**Lymphatic Vessel Infiltration**					0.093 (a)
	L0	32 (35.2%)	26 (81.3%)	6 (18.8%)	
	L1	56 (51.5%)	35 (62.5%)	21 (37.5%)	
**Venous Vessel Infiltration**					0.108 (a)
	V0	67 (73.6%)	50 (74.6%)	17 (25.4%)	
	V1	22 (24.2%)	12 (54.5%)	10 (45.5%)	
**Grading**					0.553 (b)
	G1	6 (6.6%)	4 (66.7%)	2 (33.3%)	
	G2	36 (39.6%)	28 (77.8%)	8 (22.2%)	
	G3	45 (49.5%)	29 (64.4%)	16 (25.6%)	
	G4	2 (2.2%)	1 (50%)	1 (50%)	
**Resection Status**					0.542 (a)
	R0	75 (82.4%)	53 (70.7%)	22 (20.3%)	
	R1	15 (16.5%)	9 (60%)	6 (40%)	
**Neoadjuvant Treatment**					0.031 (a)
	No	75 (82.4%)	48 (64%)	27 (36%)	
	Yes	15 (16.5%)	14 (93.3%)	1 (6.7%)	
**Recurrence**					<0.001 (a)
	No	36 (39.6%)	32 (88.9%)	4 (11.1%)	
	Yes	54 (59.3%)	30 (55.6%)	24 (44.4%)	
**CTC**					
	No	48 (73.8%)	39 (81.3%)	9 (18.8%)	<0.001 (a)
	Yes	17 (26.2%)	6 (35.3%)	11 (64.7%)	
**DTC**					
	No	49 (53.80%)	39 (79.6%)	10 (20.4%)	0.003 (a)
	Yes	14 (15.4%)	5 (35.7%)	9 (64.3%)	

**Table 2 cancers-13-04980-t002:** Subgroup analysis of esophageal squamous cell cancer patients with respect to DKK1 serum levels.

		*N* (%)	Low S-DKK1 <5800 pg/mL *N* (%)	High S-DKK1 ≥5800 pg/mL *N* (%)	*p* Value (a) Two-Sided Fisher Exact Test (b) Two-Sided Pearson Chi Quadrat Test
**Gender**					1 (a)
	Female	15 (50%)	13 (86.7%)	2 (13.3%)	
	Male	15 (50%)	13 (86.7%)	2 (13.3%)	
**Age**					0.632 (a)
	<65	12 (40%)	11 (91.7%)	1 (8.3%)	
	≥65	18 (60%)	15 (83.3%)	3 (16.7%)	
**Tumor Stage**					0.483 (b)
	T0	2 (6.7%)	2 (100%)	0 (0%)	
	T1	7 (23.3%)	7 (100%)	0 (0%)	
	T2	8 (26.7%)	7 (87.5%)	1 (12.5%)	
	T3	13 (43.3%)	10 (76.9%)	3 (23.1%)	
**Lymphatic Node Metastasis**					0.725 (b)
	N0	14 (46.7%)	12 (85.7%)	2 (14.3%)	
	N1	10 (33.33%)	8 (80%)	2 (20%)	
	N2	3 (10%)	3 (100%)	0 (0%)	
	N3	3 (10%)	3 (100%)	0 (0%)	
**Distant Metastasis**					1 (a)
	M0	29 (96.7%)	25 (86.2%)	4 (13.8%)	
	M1	1 (3.3%)	1 (100%)	0 (0%)	
**Lymphatic Vessel Infiltration**					1 (a)
	L0	16 (53.3%)	14 (87.5%)	2 (12.5%)	
	L1	14 (46.7%)	12 (85.7%)	2 (14.3%)	
**Venous Vessel Infiltration**					0.454 (a)
	V0	26 (86.7%)	23 (88.5%)	3 (11.5%)	
	V1	4 (13.3%)	3 (75%)	1 (25%)	
**Grading**					0.753 (b)
	G1	2 (6.7%)	2 (100%)	0 (0%)	
	G2	17 (56.7%)	15 (88.2%)	2 (11.8%)	
	G3	11 (36.7%)	9 (81.8%)	2 (18.2%)	
**Resection Status**					0.454 (a)
	R0	26 (86.7%)	23 (88.5%)	3 (11.5%)	
	R1	4 (13.3%)	3 (75%)	1 (25%)	
**Neoadjuvant Treatment**					0.557 (a)
	No	24 (80%)	20 (83.3%)	4 (16.7%)	
	Yes	6 (20%)	6 (100%)	0 (0%)	
**Recurrence**					0.037 (a)
	No	16 (53.3%)	16 (100%)	0 (0%)	
	Yes	14 (46.7%)	10 (71.4%)	4 (28.6%)	
**CTC**					
	No	12 (66.7%)	11 (91.7%)	1 (8.3%)	0.083 (a)
	Yes	6 (33.3%)	3 (50%)	3 (50%)	
**DTC**					
	No	15 (88.2%)	13 (86.7%)	2 (13.3%)	0.331 (a)
	Yes	2 (11.8%)	1 (50%)	1 (50%)	

**Table 3 cancers-13-04980-t003:** Subgroup analysis of esophageal adenocarcinoma patients with respect to DKK1 serum levels.

		*N* (%)	Low S-DKK1 < 5800 pg/mL *N* (%)	High S-DKK1 ≥ 5800 pg/mL *N* (%)	*p* Value (a) Two-Sided Fisher Exact Test (b) Two-Sided Pearson Chi Quadrat Test
**Gender**					0.502 (a)
	Female	11 (18%)	8 (72.7%)	3 (27.3%)	
	Male	50 (82%)	29 (58%)	21 (42%)	
**Age**					0.795 (a)
	<65	30 (49.2%)	19 (63.3%)	11 (36.7%)	
	≥65	31 (50.8%)	18 (58.1%)	13 (41.9%)	
**Tumor Stage**					0.107 (b)
	T1	14 (23%)	10 (71.4%)	4 (28.6%)	
	T2	6 (9.8%)	6 (100%)	0 (0%)	
	T3	35 (57.4%)	18 (51.4%)	17 (48.6%)	
	T4	6 (9.8%)	3 (50%)	3 (50%)	
**Lymphatic Node Metastasis**					0.146 (b)
	N0	20 (32.8%)	15 (75%)	5 (25%)	
	N1	18 (29.5%)	8 (44.4%)	10 (55.6%)	
	N2	14 (23%)	10 (71.4%)	4 (28.6%)	
	N3	9 (14.8%)	4 (44.4%)	5 (55.6%)	
**Distant Metastasis**					0.515 (a)
	M0	59 (96.7%)	35 (59.3%)	24 (40.7%)	
	M1	2 (3.3%)	2 (100%)	0 (0%)	
**Lymphatic Vessel Infiltration**					0.232 (a)
	L0	16 (27.6%)	12 (75%)	4 (25%)	
	L1	42 (72.4%)	23 (54.8%)	19 (45.2%)	
**Venous Vessel Infiltration**					0.265 (a)
	V0	41 (69.5%)	27 (65.9%)	14 (34.1%)	
	V1	18 (30.5%)	9 (50%)	9 (50%)	
**Grading**					0.846 (b)
	G1	4 (6.8%)	2 (50%)	2 (50%)	
	G2	19 (32.2%)	13 (68.4%)	6 (31.6%)	
	G3	34 (57.6%)	20 (58.8%)	14 (41.2%)	
	G4	2 (3.4%)	1 (50%)	1 (50%)	
**Resection Status**					0.741 (a)
	R0	49 (81.7%)	30 (61.2%)	19 (38.8%)	
	R1	11 (18.3%)	6 (54.5%)	5 (45.5%)	
**Neoadjuvant Treatment**					0.072 (a)
	No	51 (85%)	28 (54.9%)	23 (45.1%)	
	Yes	9 (15%)	8 (88.9%)	1 (11.1%)	
**Recurrence**					0.029 (a)
	No	20 (33.3%)	16 (80%)	4 (20%)	
	Yes	40 (66.7%)	20 (50%)	20 (50%)	
**CTC**					
	No	36 (76.6%)	28 (77.8%)	8 (22.2%)	0.004 (a)
	Yes	11 (23.4%)	3 (27.3%)	8 (72.7%)	
**DTC**					
	No	34 (73.9%)	26 (76.5%)	8 (23.5%)	0.013 (a)
	Yes	12 (26.1%)	4 (33.3%)	8 (66.7%)	

**Table 4 cancers-13-04980-t004:** Univariate analysis of cancer recurrence in esophageal cancer patients with respect to S-DKK1, presence of CTCs and DTC detection.

		*N*	Time to Relapse (Median) in Months	(95% CI) ± Standard Error	Logrank Test (Mantel-Cox) *p*-Value
**S-DKK1**					<0.0005
	Low	62	22	(15.43–28.57) ± 3.35	
	High	28	12	(8.2–15.8) ± 1.94	
**CTC**					0.003
	No	48	22	(12.43–31.57) ± 4.883	
	Yes	17	11	(9.01–12.99) ± 1.01	
**DTC**					0.059
	No	49	18	(8.3–27.70) ± 4.95	
	Yes	14	11	(7.33–14.67) ± 1.87	

**Table 5 cancers-13-04980-t005:** Univariate analysis of overall survival in esophageal cancer patients with respect to S-DKK1, presence of CTCs and DTC detection.

		N	Overall Survival (Median) in Months	(95% CI) ± Standard Error	Logrank (Mantel-Cox) *p*-Value
**DKK1**					0.003
	low	63	26	(21.30–30.70) ± 2.40	
	high	28	14	(8.65–19.36) ± 2.73	
**CTC**					0.010
	No	48	26	(21.47–30.53) ± 2.31	
	Yes	17	13	(6.59–19.41) ± 3.27	
**DTC**					0.116
	No	49	24	(17.58–30.42) ± 3.28	
	Yes	14	13	(5.67–20.33) ± 3.74	

## Data Availability

All data generated during the study are included in this manuscript.

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
