# Peer review of "High Serum Levels of Wnt Signaling Antagonist Dickkopf-Related Protein 1 Are Associated with Impaired Overall Survival and Recurrence in Esophageal Cancer Patients"

_cancers, 2021, doi:10.3390/cancers13194980_

Round 1

Reviewer 1 Report

The research article submitted by Ramirez et al on 30 patients with esophageal squamous cell cancer and 61 esophageal adenocarcinoma patients showed that higher levels of DKK1 expression were associated with a worse prognosis in these  patients. Overall, this is a timely paper for the field. The organization is of the context is clear and easy to follow. Therefore, it should be published given some minor issues stated in the following.

The following are a few comments/suggestions:

As the author appropriately point out, multivariate analysis identified S-DKK1 as an independent prognostic marker with respect to overall survival in EC patients. Yet more discussion should be dedicated to how these findings can be translated into clinical applications in the future. We believe this issue  represents a great deal of interests of the readers.

Author Response

Reviewer 1:

The research article submitted by Ramirez et al on 30 patients with esophageal squamous cell cancer and 61 esophageal adenocarcinoma patients showed that higher levels of DKK1 expression were associated with a worse prognosis in these patients. Overall, this is a timely paper for the field. The organization is of the context is clear and easy to follow. Therefore, it should be published given some minor issues stated in the following.

RE: Dear reviewer 1, we would like to thank you for the positive comments on our manuscript. Please find below our response to your suggestions. We hope that we could address your concerns to satisfaction in this revised version. Thank you and best regards.

The following are a few comments/suggestions:

As the author appropriately point out, multivariate analysis identified S-DKK1 as an independent prognostic marker with respect to overall survival in EC patients. Yet more discussion should be dedicated to how these findings can be translated into clinical applications in the future. We believe this issue represents a great deal of interests of the readers.

RE: We strongly agree that prognostic markers including DKK1 should be implemented into the clinical routine. Therefore, we revised the discussion and added potential clinical applications of DKK1 in greater detail.

Page 15, line 343:

“Another potential application may be S-DKK1 as a novel treatment target. Targeted therapies in EC have recently been established against EC with distinct molecular signatures [30]. For example, trastuzumab, a monoclonal antibody targeting human epi-dermal growth factor receptor 2 (HER2) in combination with chemotherapy, has recently been approved as first line therapy by the FDA for EC patients [30]. This drug targets the constitutive activation of several pathways including the oncogenic RAS/RAF/MEK/ERK pathway which is subsequently activated by amplification of HER2 [31]. However, as discussed above, the functional role of DKK1 is still unknown and therefore targeted therapy inhibiting DKK1 should be carefully evaluated in further experiments in vitro as well as xenograft models in vivo. Recently, Lyros et al. reported that DKK1 promotes tumor growth by attenuating the PI3K/AKT/mTOR axis independently of Wnt signaling in EC OE33 cells in vitro [32]. While according to this study an inhibition of DKK1 would be beneficial, in other studies including breast cancer a tumor-suppressive effect of DKK1 with respect to migration and invasion was observed [33]. In our study, we found out that high levels of S-DKK1 are associated with poor prognosis. Therefore, the data are rather in line with a tumorigenic role of DKK-1, although further molecular analysis is necessary to confirm this idea.“

Page 16, line 368:

“In addition to treatment choice and the potential use as a treatment target, the use of S-DKK1 in EC may also facilitate identification of high-risk patients that would benefit from intensified aftercare (e.g., in the form of additional clinical imaging and shorter consultation intervals). Recent research demonstrated that in lung cancer especially the combination of blood-based biomarkers and CTC detection is superior to use of single biomarkers [35]. Therefore, we think that the combination of CTCs and S-DKK1 levels may be useful for a more personalized cancer therapy.“

Reviewer 2 Report

The authors showed that high serum levels of DKK1 were associated with poor prognosis and recurrence in patients of esophageal cancer (EC) including adenocarcinoma (EC) and squamous cell carcinoma (ESCC).

I have some questions.

  1. Did the expression levels of S-DKK1 correlate with expression levels of DKK1 in the primary tumor?
  2. Was S-DKK1 more useful biomarker than other tumor markers of EC such as SCC or CEA?
  3. Why did you choose obese patients as a control?

Author Response

Reviewer 2:

The authors showed that high serum levels of DKK1 were associated with poor prognosis and recurrence in patients of esophageal cancer (EC) including adenocarcinoma (EC) and squamous cell carcinoma (ESCC).

RE: Dear reviewer 2, thank you very much for your feedback on our manuscript. Please find below our point-to-point response to the questions you raised. We hope that we were able to answer them to your satisfaction. Thank you and best regards.

 I have some questions.

1) Did the expression levels of S-DKK1 correlate with expression levels of DKK1 in the primary tumor?

RE: Unfortunately, we did not analyze tumor tissues for DKK1 expression as those were not collected in this project. However, in a previous study by Han et al. investigating the role of DKK1 in prostate cancer, the authors were able to detect a strong positive correlation between high S-DKK1 levels and strong DKK1 expression in primary tumor samples (Han et al., Oncotarget. 2015 Aug 14;6(23):19907-17).

As we think that this correlation is of interest to the reader, we added this reference to our discussion.

Page 16, line 361:

“In primary prostate cancer, a strong positive correlation of S-DKK1 levels and tissue DKK1 expression levels within the tumor was reported in the past [34].”

2) Was S-DKK1 more useful biomarker than other tumor markers of EC such as SCC or CEA?

RE: This is an interesting point, however, as those tumor markers are not validated for esophageal cancer and not suggested in the current clinical practice guidelines in Germany, we did not analyze those markers on a regular base.

As we think that this is an important point, we extended our discussion based on a recent meta-analysis that evaluated biomarkers in EC. The authors concluded that due to the low sensitivity, CEA, Cyfra21–1, p53 antibody, SCC-Ag and VEGF-C should not be used alone for diagnosis of EC (Zhang et al. PLOS ONE 10(2): 2015 e0116951). Therefore, we propose that a combination of biomarkers including S-DKK1 may be a useful tool in EC.

Page 15, line 330:

“Despite extensive research in the past, in contrast to other gastrointestinal malignancies including colorectal cancer, no established tumor markers for esophageal cancer exist yet [28]. Nevertheless, a meta-analysis by Zhang et al. demonstrated that tumor markers including carcinoembryonic antigen, cytokeratin-19 fragment 21-1, p53 anti-body, squamous cell carcinoma antigen and VEGF-C are highly specific but lack the required sensitivity to precisely diagnose EC [29]. Therefore, multiple biomarkers including S-DKK1 may be combined to increase sensitivity.”

3) Why did you choose obese patients as a control?

RE: We used obese patients as a control, as they were required to have a recent esophagogastroduodenoscopy (< 6 months) prior to bariatric surgery. Thereby we ensured that no esophageal carcinoma was present in patients within the control group.

Reviewer 3 Report

The manuscirpt by Ramirez et al. presented DKK1 as a prognostic marker in esophageal cancer patients. They showed that elevated DKK1 levels are associated with low survival and early recurrence in patients with such cancer types. The study was well designed and the conclusion was solid.

The biological roles of DKK1 in esophageal cancers are still ambiguous. Does DKK1 promote cancer progression on its own? Or the high expression levels of DKK1 are induced by Wnt signaling to form a negative feedback loop? In the latter case, could it be that the high Wnt activation is causing the poor prognosis and increased DKK1 levels are just a secondary effect? Then, targeted inhibition of DKK1 will not be a wise choice for the patients. Maybe also check the Wnt signal activity (beta-catenin nuclear translocation) and then determine whether to target DKK1? Will be great to see one section in the discussion to talk about the above points.

Author Response

Reviewer 3:

The manuscirpt by Ramirez et al. presented DKK1 as a prognostic marker in esophageal cancer patients. They showed that elevated DKK1 levels are associated with low survival and early recurrence in patients with such cancer types. The study was well designed and the conclusion was solid.

RE: Dear reviewer 3, thank you very much for the positive evaluation of our manuscript. Please find our reply to your suggestion below. Thank you and best regards. 

The biological roles of DKK1 in esophageal cancers are still ambiguous. Does DKK1 promote cancer progression on its own? Or the high expression levels of DKK1 are induced by Wnt signaling to form a negative feedback loop? In the latter case, could it be that the high Wnt activation is causing the poor prognosis and increased DKK1 levels are just a secondary effect? Then, targeted inhibition of DKK1 will not be a wise choice for the patients. Maybe also check the Wnt signal activity (beta-catenin nuclear translocation) and then determine whether to target DKK1? Will be great to see one section in the discussion to talk about the above points.

RE: Thank you very much for pointing this out. We highly agree that it would be an interesting experiment to use for example membrane fractioning to separate cytosolic from nuclear proteins. However, as this manuscript mainly focuses clinical aspects and risk stratification, and the experiments would need several months to complete we think that this would be beyond the scope of the current manuscript. Nevertheless, we agree and will definitely consider this experiment in upcoming work. To account for your comment, we extended the discussion. It now reads:

Page 15, line 351:

“However, as discussed above, the functional role of DKK1 is still unknown and therefore targeted therapy inhibiting DKK1 should be carefully evaluated in further experiments in vitro as well as xenograft models in vivo. Recently, Lyros et al. reported that DKK1 promotes tumor growth by attenuating the PI3K/AKT/mTOR axis independently of Wnt signaling in EC OE33 cells in vitro [32]. While according to this study an inhibition of DKK1 would be beneficial, in other studies including breast cancer a tumor-suppressive effect of DKK1 with respect to migration and invasion was observed [33]. In our study, we found out that high levels of S-DKK1 are associated with poor prognosis. Therefore, the data are rather in line with a tumorigenic role of DKK-1, although further molecular analysis is necessary to confirm this idea. In primary prostate cancer, a strong positive correlation of S-DKK1 levels and tissue DKK1 expression levels within the tumor was reported in the past [34]. Nevertheless, it remains unclear whether the high S-DKK1 levels are a direct tumorigenic stimulus that promotes tumor growth and malignancy or whether they are induced by a negative feedback loop as a sign of constitutive Wnt signaling. Hence, further molecular characterization of the underlaying mechanism is needed to provide a robust answer to the question whether DKK1 may be a suitable target in EC.“

Round 2

Reviewer 2 Report

I am satisfied with the author's responses to my comments.